# Determining the Degree of Admixing Rate of the Base Material and the Melting Efficiency in Single-Bead Surface Welds Using Different Methods, Including New Approaches

**DOI:** 10.3390/ma12091479

**Published:** 2019-05-07

**Authors:** Matija Zorc, Aleš Nagode, Borut Kosec, Borut Zorc

**Affiliations:** 1Faculty of Natural Sciences and Engineering, University of Ljubljana, Aškerčeva 12, 1000 Ljubljana, Slovenia; matija.zorc@omm.ntf.uni-lj.si (M.Z.); ales.nagode@omm.ntf.uni-lj.si (A.N.); borut.kosec@omm.ntf.uni-lj.si (B.K.); 2Welding Institute Ltd., Ptujska 19, 1000 Ljubljana, Slovenia

**Keywords:** arc welding, single-bead surface weld, admixing rate, melting efficiency

## Abstract

The precise determination of the admixing rate of the base material for certain welding parameters is very important because of the possible negative consequences. As such, it is the basis for corrections in welding technology. In the article, experimental and theoretical determinations of the admixing rate in single-bead surface welds that were arc welded onto S355 steel with different alloyed-steel-coated electrodes are discussed. The admixing rate was experimentally estimated from the ratio of the surface areas of metallographic cross-sections, from the ratios of the height and from chemical analyses of different regions of the surface weld, while it was theoretically estimated from the characteristics of the welding process and material constants. One of the key characteristics of the welding process is the melting efficiency, which can be estimated by means of different equations and from knowledge of the heat balance of the welding process. Both the average melting efficiency of the surface welding on the medium-thick S355 steel plate and the average admixing rate of the S355 steel into the surface welds have the same value, i.e., approximately 30%. New equations for estimating the melting efficiency of the arc welding with a coated electrode were developed on the basis of the results.

## 1. Introduction

During fusion welding, a part of the melted base material, which usually amounts to 10–40% when welding with coated electrodes [1,2], is admixed into the filler material melt [1,2,3,4,5,6,7,8,9,10,11]. The amount of admixed base material in a single-bead surface weld is determined, e.g., on the basis of the surface areas of the weld from the metallographic section. When the degree of admixed base material is known, the approximate amount of all the alloying elements in the mixing region can be calculated [3,5]. Because the melt is being intensively mixed as a result of the various forces that act during the arc welding, the mass transport is much faster than with diffusion in the solid state. The mixing of the melt during welding with coated electrodes is further accelerated by the molten drops that were falling into the weld pool. Despite a very intensive mixing of the melt, the holding time of the weld pool in the liquid state is too short, with the consequence being the presence of a chemical heterogeneity in the form of macro- and micro-segregations in the weld.

The macro-segregations in the weld can be indicated by etching and they are visible in the form of darker or lighter lines that can be seen at lower magnifications or even with the naked eye. Macro-segregations are directly related to the mixing of the filler and the base material, and are more strongly expressed when welding two different materials [5,9]. In the case of fusion welding without any filler material, the amount of macro-segregations in the welds is especially large for high-energy welding processes, e.g., laser-beam welding [5].

There are three types of micro-segregations in welds. The first type is not dependent on the admixing of the base material. It is related to the segregation coefficient of the alloying element and is a consequence of the non-equilibrium solidification of the whole volume of the weld pool, which is shown in the form of both positive and negative segregations of the alloying elements at, or on, the grain boundaries of the trans-crystalline dendrites [1,12]. The second type of micro-segregation is a band formation during the solidification of the weld pool. This type of micro-segregation is related neither with the segregation coefficient of the alloying elements nor with the admixing of the base material into the weld, but is a consequence of the specific conditions that are characteristic for welding processes [5]. The third type of micro-segregation in a weld is a partially mixed zone that exists near the fusion line and is conditioned by a different content of alloying elements in both the filler and the base materials. This type of micro-segregation is a direct consequence of the base material admixing into the weld. The larger is the fraction of melted base material, the wider is this area. Its width is generally from 50 μm to a few hundred microns [4,5,6,7,8,13,14]. There was even a completely unmixed layer found close to the fusion line in the case of the welding of austenitic stainless steel, which consisted only of the melted base material [5,15]. This indicates a negligible mixing of the thin layer of the melt that is in contact with the solid base material [13,16]. In practice, when welding two different materials together it is preferable to have as little admixing of the base material into the weld as possible, because it can lead to the formation of an unfavourable brittle microstructure in certain areas and consequently to a higher sensitivity to cracking in these areas.

The article presents a comparative analysis of experimental and theoretical methods for determining the admixing rate of the base material in single-bead surface welds that were arc surface welded onto S355 low-alloyed steel with different alloyed coated electrodes. Using different equations, the article also includes a theoretical estimation of the melting efficiency of the welding process, which is necessary for a theoretical evaluation of the degree of admixing.

## 2. Materials and Methods

### 2.1. Preparation of the Materials and Samples

Eight steels with different chromium contents (see Section 3.3.) were manually arc welded (welding power source “Fronius magic waves 3000”) onto the surface of a 20-mm-thick and 500 mm × 500 mm non-preheated S355 low-alloyed steel sheet in a flat position (PA) with commercial, basic coated electrodes for the welding of common construction steel (weld no. 1), for the welding of creep-resistant steel (welds no. 2–4) and for surfacing (welds no. 5–8). The diameter of the electrodes was 3.25 mm. The welds were single-bead and 150 mm long. The welding was carried out with direct current, having the (+) pole on the electrodes. The welding parameters were current *I* = 100 ± 1.5 A, voltage *U* = 22.5 ± 3 V, and welding speed *v_w_* = 2.83 ± 0.35 mm/s, while the arc efficiency is a standardized value *η_a_* = 0.8 [17]. The average heat input of a single weld was *Q* = (*U·I·η_a_*)/*v_w_* = 636 J/mm.

As the mixing of the melt during arc welding is very intense, it is not possible to determine the chemical composition of the filler metal from a single-bead surface weld. Therefore, for the determination of the chemical composition of the filler metals alone, three-layered surface welds were made with the same welding parameters (the first layer with five beads, the second layer with four beads and the third layer with three beads).

For the analyses, a single cross-section was cut in the middle of each surface weld with rapid cooling of the samples. Each cross-section was wet-ground with #80 to #1000 SiC papers. Etching with 10% Nital was performed, which made the single-bead welds visible; therefore, it was possible to determine the degree of admixing of the S355 steel into the single-bead surface weld. This also revealed the microstructure of the less-alloyed surface welds, while in the more-alloyed surface welds only the microstructure of the S355 steel was visible. The etched metallographic samples were photographed using a macro lens.

### 2.2. Research Methods

The chemical analyses of the surface of the three-layered weld in order to determine the chemical composition of the filler metal (analyses were made at two positions) and the chemical analyses of the metallographic cross-sections of the single-bead surface welds in order to determine the admixing rate (four measurements in the area of the filler material and two measurements in the area of the base material) were made using a JEOL JSM-5610 SEM (JEOL, Tokyo, Japan) with Energy-dispersive x-ray spectrometer—EDS (Gresham Scientific Instruments, London, UK). In the S355 steel and in surface weld no. 1 the controlled chemical elements were Si and Mn, while in the other surface welds it was Cr (its content in S355 steel and surface weld no. 1 is negligible).

There are several ways to determine the admixing rate of the base material in the surface weld described in the literature: from the areas in the metallographic cross-section of the weld [3,5,6,7,8,9,10], from the mass of the melted filler and base materials [4], from the volume fraction of the melted filler and base materials [6,7], from the chemical composition measured by an electron-probe micro-analyser [7] and from a calculation using the welding parameters, the arc and melting efficiency and the melting enthalpy of the base and filler materials [6,7]. In our case, the admixing rate of the base material into the single-bead surface welds was determined in the following ways:-From the ratio of the areas in the metallographic cross-section, which in a single-bead surface weld belong to the base material and to the filler material; the areas were measured with the “imageJ” computer program.-From the ratio of the heights on the metallographic cross-section, which in a single-bead surface weld belong to the base material and to the filler material (a new approach); the heights were measured with the “imageJ” computer program.-By calculating from the chemical composition of the filler metal and the chemical composition of the areas, which in a single-bead surface weld belong to the filler material (four measurements) and to the base material (two measurements).-By calculating from the welding parameters and the material constants.

The experimental methods are shown in Figure 1.

## 3. Results and Discussion

### 3.1. Admixing Rate from the Ratio of the Areas

Based on the areas in the metallographic cross-section (Figure 1a) the admixing rate of the base material in the single-bead surface weld *D_A_* was calculated using the equation [3,6,7,8,9,10]:
(1)$$D_{A}=\frac{A_{bm}}{A_{bm}+A_{fm}}\times 100=\frac{A_{bm}}{A_{w}}\times 100,\;(\%)$$
where *A_bm_* is the area of the surface weld that belongs to the base material (mm^2^), *A_fm_* is the area of the surface weld that belongs to the filler material (mm^2^) and *A_w_* is the whole area of the surface weld (mm^2^). The measured areas *A_bm_* and *A_w_* and the admixing rate *D_A_* are given in Table 1.

The values determined for the admixing rate of the S355 steel into the individual surface welds are from 21% to 38.8% (average *D_A_* = 30.3%) and are comparable with the values of the arc welding with coated electrodes found in the literature. The differences in the degree of admixing for different beads (even by 18%) are a consequence of the manual welding (fluctuations of the welding speed and the arc length), the different coatings and the characteristics of the different electrode types.

The degree of admixing determined from the areas *D_A_* was also controlled with a calculation of the average chromium content of Cr*^bc^* (in the non-alloyed surface weld no. 1, Si*^bc^* and Mn*^bc^* were calculated), which is described in Section 3.3.

### 3.2. Admixing Rate from the Ratio of the Heights

This method was not found in any literature available to us and represents a new approach to determining the admixing rate of the base material into the surface weld. From the metallographic cross-sections, it is clear that the shape of the fusion line and the shape of the top of the surface weld are similar when arc welding with a coated electrode. When idealized, the areas represent two circular segments: the area of the base metal *A_bm_* from the circle with the larger radius and the area of the filler metal *A_fm_* from the circle with the smaller radius. In an idealized symmetrical shape, both cross-sections are the largest in the middle of the surface weld. The ratio of the height of the sunken part of surface weld *h_bm_* and the height of the whole surface weld *h_w_* on a chosen line represents the admixing rate of the base material on this line. As the ratio changes towards the edge of the surface weld and it can also fluctuate slightly, it is necessary to perform several measurements on several vertical lines from the middle towards the edges of the surface weld in order to determine the partial admixing rates and to calculate an average for these values (seven vertical lines were analysed in each surface weld in our case, Figure 1b). The admixing rate of the base material *D_h_* in one surface weld from its heights is calculated using the equation:
(2)$$D_{h}=\frac{\sum_{i=1}^{n}h_{bm}}{\sum_{i=1}^{n}\left(h_{bm}+h_{fm}\right)}\times 100=\frac{\sum_{i=1}^{n}h_{bm}}{\sum_{1=1}^{n}h_{w}}\times 100,\;(\%)$$
where *h_bm_* is the height of the segment of the surface weld that belongs to the base material (mm), *h_fm_* is the height of the segment of the surface weld that belongs to the filler material (mm) and *h_w_* is the whole height of the surface weld (mm). The measured heights *h_bm_* and *h_w_* and the calculated values of the admixing rate *D_h_* are given in Table 2.

The values determined for the admixing rate of the S355 steel into the individual surface welds are from 21.1% to 37.9%, while the average value of all the measurement is *D_h_* = 30.2%. The results show a good matching of the individual values of *D_h_* and *D_A_*, while the average value of all the measurements is practically the same. The results show that measurements of the appropriate heights on a large enough number of appropriately chosen vertical lines in the individual cross-sections are the guarantee of an accurate determination of the admixing rate from the ratio of the heights.

### 3.3. Admixing Rate from the Chemical Composition of the Surface Weld

The admixing rate from the chemical composition *D_ch_* was calculated from the Cr content in the filler metal (marked *a* next to the number of the surface weld in Table 3), from the average value of all six measured values of Cr in each cross-section (marked with *ev* next to the number of the surface weld in Table 3) and from the average value of the two lowest measured values of Cr in each cross-section (marked with *min* next to the number of the surface weld in Table 3).

The average admixing rate from all the measured values *D^t^_ch_* and from the two lowest measured values *D^l^_ch_* in the individual welds can be calculated using the equations:
(3a)$$D_{ch}^{t}=\left(1-\frac{Cr^{ev}}{Cr^{a}}\right)\times 100;\;(\%)$$
(3b)$$D_{ch}^{l}=\left(1-\frac{Cr^{\min}}{Cr^{a}}\right)\times 100,\;(\%)$$

For example, for the surface weld no. 4 from Table 3:$$D_{ch}^{t}=\left(1-\frac{6.53}{9.32}\right)\times 100=29.9\%;\;D_{ch}^{l}=\left(1-\frac{6.455}{9.32}\right)\times 100=30.7\%$$

The comparison of the values of *D_ch_* and *D_A_* is evident in Table 3. We have 14 values of *D_ch_* (7 values of *D^t^_ch_* and 7 values of *D^l^_ch_*). The results show that the *D_ch_* and *D_A_* values match well (9 of the 14 *D_ch_* values match *D_A_* with a 90% to 99% accuracy, three values with an 81% to 88% accuracy and two values with a 74% to 78% accuracy). The slightly larger deviations are a consequence of the manual welding (both the length of the arc and the welding speed change slightly during the process) and due to this, a fluctuation in the amount of admixed base material occurred. As the radial gradient of the arc pressure is formed between the front and the back sides of the weld pool due to the straight movement of the electrode [18], the melt flows from the front side towards the back side of the weld pool. Therefore, the admixing rate that is determined from the chemical composition of the analysed metallographic cross-section does not match the admixing rate that is determined from the areas. The metallographically determined admixing rate, i.e., *D_A_*, lags behind the chemical *D_ch_* by a certain distance, which is dependent on the length of the weld pool (the chemical composition measured on the analysed cross-section actually belongs to the cross-section that is forward by a certain distance in the direction of welding). Therefore, the *D_ch_*/*D_A_* > 1 ratio means that in the mentioned section, the welder had a somewhat shorter arc than in the analysed cross-section. Because of that, the amount of back flow and the admixing of the S355 steel is larger. Meanwhile, a ratio *D_ch_*/*D_A_* < 1 means that in this section the welder had a slightly longer arc than in the analysed cross-section. Therefore, the amount of back flow and admixing of the S355 steel is lower. Despite the slightly larger deviations for the chemically determined admixing *D_ch_* in the individual surface welds in comparison with *D_A_*, the average degree of the seventh value of admixing determined from all the measurements in the individual cross-sections (values marked with *ev* in Table 3) is ∑ *D^t^_ch_* = 29% and the average degree of the seventh value of admixing from two lowest measurements in an individual cross-section (values marked with *min* in Table 3) is ∑ *D^l^_ch_* = 32%. If we take into account all 14 values or *D_ch_* = (∑ *D^t^_ch_* + ∑ *D^l^_ch_*)/2, the chemically determined admixing rate is *D_ch_* = 30.5%, which is practically the same as the degree determined using the metallographic methods. The results show that for each cross-section it is especially useful to calculate the admixing rate from all the measured values *D^t^_ch_* and from the two lowest values *D^l^_ch_*, and finally to calculate the average of all these values. This gives the possibility for a very accurate determination of the admixing rate from the chemical composition.

The admixing rate *D_A_* of each cross-section was controlled with a calculation of the average content of chromium Cr*^bc^*, which was compared to two average values of chromium: the average of all six measured values (marked with *ev* at the number of the weld) and the average of the two lowest measured values (marked with *min* at the number of the weld), as in Table 3. The content of Cr*^bc^* is calculated from the chromium content in the filler metal Cr*^a^* using the equation:(4)$$Cr^{bc}=Cr^{a}\times \left(1-0.01D_{A}\right),\;(\mathrm{wt}.\%)$$

For example, for surface weld no. 2: $Cr^{bc}=1.03\times \left(1-0.359\right)=0.66$ wt.%.

The admixing rate in surface weld no. 1, *D_A_* was controlled with Si and Mn. As both alloying elements exist in both components, the equation for the calculation of the values Si*^bc^* and Mn*^bc^* in the surface weld is:(5)$$X^{bc}=\left[X^{S355}\times 0.01D_{A}\right]+\left[X^{a}\times \left(1-0.01D_{A}\right)\right],\;(\mathrm{wt}.\%)$$

*X*^*S*355^ is the content of the chosen element in the S355 steel and *X^a^* is the content of that same element in the filler metal. Thus, for silicon: $Si^{bc}=\left[0.42\times 0.337\right]+\left[0.65\times \left(1-0.337\right)\right]=0.57$ wt.%, while for manganese: $Mn^{bc}=\left[1.00\times 0.337\right]+\left[0.9\times \left(1-0.337\right)\right]=0.93$ wt.%.

The comparison of the calculated and measured values shows a high degree of matching (from 90% to 99.7% or 96%, on average), which means that the admixing rate can be correctly estimated for the average chemical composition of the surface weld. The results also show that the degree of mixing is valid for the whole volume of the single-bead surface weld that was made with the coated electrode.

### 3.4. Summary of the Experimental Determination of the Admixing

Table 4 presents the admixing rate determined by different experimental methods. From the results, we can conclude that despite little or more deviation of the individual values for the same cross-section, the average admixing rate, determined from eight cross-sections with different methods, is practically the same: *D_A_* = 30.3%, *D_h_* = 30.2%, *D_ch_* = 30.5%. This means that all the tested methods are equivalent for the determination of the average admixing rate for a sufficiently large number of cross-sections. Therefore, the average admixing rate of the S355 steel into the surface welds in our case is *D_m_* = 30.3% ≈ 30%. The results show that an accurate determination of the average admixing rate in an individual weld is possible by analysing several metallographic cross-sections, taken from the entire length of the weld at various distances from the start to the end of the weld.

### 3.5. Theoretical Determination of the Admixing Rate from the Welding Parameters

The degree of base material admixing *D_cal_* can be calculated on the basis of the welding parameters and the physical constants of the welded material using the equation [6,7]:(6)$$D_{cal}={\left(1+\frac{V_{fm}\cdot E_{bm}}{\eta_{a}\cdot \eta_{m}\cdot U\cdot I-E_{fm}\cdot V_{fm}}\right)}^{-1}\times 100,\;(\%)$$
where *V_fm_* is the quantity of the melting filler material (mm^3^/s), *E_bm_* and *E_fm_* are the melting enthalpies of the base and filler materials (J/mm^3^), *η_a_* and *η_m_* are the arc and melting efficiency, *U* is the voltage (V) and *I* is the current (A).

The main problem with Equation (6) is the determination of the melting efficiency *η_m_*, which is dependent on the welding parameters, the dimensions of the surface weld and the physical properties of the material, as well as its thickness [6].

Several equations for the calculation of *η_m_* exist. Some are applicable for three-dimensional (3D) or two-dimensional (2D) heat flows, while others are not dependent on the type of heat flow. The type of heat flow is determined by calculating the relative thickness of the welded material τ [19]:(7)$$\tau =t\cdot \sqrt{\frac{C\cdot \rho \cdot \left(T_{c}-T_{0}\right)}{U\cdot I\cdot \eta_{a}}}$$
where *t* is the thickness of the welded material (mm), *C∙ρ* is the volumetric heat capacity (J/(mm^3^∙°C)), *T_c_* is the temperature at which the cooling speed is calculated (°C), and *T*_0_ is the initial temperature of the welded material (°C). For other symbols, refer to Equation (6).

When simplified, the 3D heat flow (thick plate) is at *τ* > 0.75, while the 2D heat flow (thin plate) is at *τ* < 0.75 (more precisely, the 3D heat flow is at *τ* > 0.9 and the 2D heat flow is at *τ* < 0.6, while between these values there is a medium-thick plate). This simplification, when calculating the cooling speed and the preheating temperature, causes an error smaller than 15% [19]. Therefore, the simplified version is generally used. If the typical values for steel (the mean value for the non-alloyed and low-alloyed steels: *C∙ρ* = 5.05 × 10^−3^ J/(mm^3^∙°C) [20,21], *T_c_* = 550 °C [19]) and our average welding conditions (*t* = 20 mm, *T*_0_ = 20 °C, *U* = 22.5 V, *I* = 100 A, *η_a_* = 0.8) are inserted into the equation, the result is *τ* = 0.77. Despite the fact that the result of the simplified version is 3D heat flow, it is clear that we are dealing with a medium-thick plate as the result of the equation is very close to the limit *τ* = 0.75. For this reason, all the equations for the calculation of the melting efficiency were considered, regardless of the type of heat flow.

#### 3.5.1. Determination of the Melting Efficiency from the Welding Parameters

For 3D heat flow during the melting of the surface of the thick plate without filler material (the shape of the surface weld is a half cylinder with a depth-to-width ratio *h_bm_*/*b_w_* = 0.5), a theoretical maximum value for the melting efficiency *η_m_* = 0.386 is cited in [20,21] or 0.37 cited in [6]. The diagram (Figure 134 in [20] and Figure 3.8 in [21]) shows that the melting efficiency is better in a shallower surface weld (for the ratio *h_bm_*/*b_w_* = 0.1 it can reach a maximum value of 0.46) and that it becomes worse with lower welding speeds, as also mentioned in [7].

For 2D heat flow, there is a maximum theoretical melting efficiency *η_m_* = 0.484 cited in the literature [6,20,21]. This is graphically shown in Figure 135 in [20] and in Figure 3.9 in [21]. In both diagrams the melting efficiency is graphically determined on the basis of dimensionless factors:
(8)$$\xi^{3D}=\frac{U\cdot I\cdot \eta_{a}\cdot v_{w}}{\alpha^{2}\cdot E_{bm}}$$
(9)$$\xi^{2D}=\frac{U\cdot I\cdot \eta_{a}}{\alpha \cdot t\cdot E_{bm}}$$

There are several equations for a direct calculation of the melting efficiency *η_m_* in the literature [6,21,22,23,24,25,26,27]:
(10)$$\eta_{m}^{2D}=\frac{1}{\frac{8\alpha }{5v_{w}\cdot b_{w}}+2},$$
(11)$$\eta_{m}^{3D}=\frac{1}{\frac{e}{2}\cdot \left[1+{\left(1+\frac{10.4\cdot \alpha^{2}}{v_{w}^{2}\cdot b_{w}^{2}}\right)}^{\frac{1}{2}}\right]},$$
(12)$$\eta_{m}=\exp\left(-1-\frac{\alpha^{2}\cdot E}{1.14\cdot \eta_{a}\cdot U\cdot I\cdot v_{w}}\right),$$
(13)$$\eta_{m}=\eta_{\max}\cdot \exp\left(\frac{\psi \cdot E\cdot \nu \cdot \alpha }{U\cdot I\cdot \eta_{a}\cdot v_{w}}\right),$$
(13a)$$\eta_{m}=0.5\cdot \exp\left(\frac{-175\cdot E\cdot \nu \cdot \alpha }{U\cdot I\cdot \eta_{a}\cdot v_{w}}\right),$$
(13b)$$\eta_{m}^{2D}=0.41\cdot \exp\left(\frac{-29.6\cdot E\cdot \nu \cdot \alpha }{U\cdot I\cdot \eta_{a}\cdot v_{w}}\right),$$
(13c)$$\eta_{m}^{3D}=0.346\cdot \exp\left(\frac{-0.9\cdot E\cdot \nu \cdot \alpha }{U\cdot I\cdot \eta_{a}\cdot v_{w}}\right),$$
(14)$$\eta_{m}=\frac{E\cdot A_{w}\cdot v_{w}}{\eta_{a}\cdot U\cdot I},$$
(15)$$\eta_{m}=\frac{M\cdot E}{U\cdot I\cdot \eta_{a}},$$
(16)$$\eta_{m}=0.065+0.016\cdot \left(\frac{E\cdot \alpha^{2}}{U\cdot I\cdot \eta_{a}\cdot v_{w}}\right),$$

In Equations (8) to (16): *E* is the average value of the melting enthalpy of the base and filler materials (J/mm^3^) or in (J/g) in Equation (15); *α* is the thermal diffusivity of the base material at 20 °C (mm^2^/s); *M* is the total mass of the melted material (g/s); *v_w_* is the welding speed (mm/s); *b_w_* is the width of the weld (mm); *A_w_* is the cross-section of the weld (mm^2^); ν is the kinematic viscosity at the melting temperature (mm^2^/s); *η_max_* is the maximum theoretical melting efficiency for a chosen shape of joint or geometry of the substrate; *ψ* is a constant. For other symbols, refer to Equation (6).

It is known that the calculated value of the melting efficiency can differ due to different equations [23]. The reason is in the specificity of the experimental conditions and in the applied welding processes. The listed equations were tested using our conditions of arc surface welding with basic coated electrodes, while the average value *η_m_* was checked using Equation (6). The following values were tried as well, when calculating the degree of admixing and the melting efficiency using these equations. The enthalpies are: *E_bm_* = 10.5 J/mm^3^ [6,7], *E_fm_* = 10.0 J/mm^3^ (as Cr at contents < 10 wt.% minimally lowers the melting temperature of iron, the enthalpy does not differ significantly), *E* = 10.25 J/mm^2^ or in Equation (15) *E* = 1340 J/g [27]. The quantity of the melting filler material *V_fm_* was calculated from Table IV-3 in [28] and from Figure 5b in [29], where it is evident that the quantity of melted coated electrodes is approximately the same, regardless of the type of coating, and that it amounts to 17.6 g/min when using a 3.2-mm-diameter electrode and a welding current *I* = 110 A. As our surface welds were made with *I* = 100 A, a value 17.0 g/min = 0.283 g/s was assumed. As the *m* = 0.283 g = *ρ_Fe_*·*V* = (0.00785 g/mm^3^)·*V*, it means that *V_fm_* = 0.283/0.00785 = 36 mm^3^/s. The total mass *M* of melted material was also calculated. From the determined admixing rate *D_m_* = 30%, it can be concluded that 0.283 g/s belong to 70% of our average surface weld. Therefore, it is necessary to add the part of the melted base material where the same amount of melted material in g/s can be assumed as for the electrode. Because the melted base material portion is 0.283 × 0.3 = 0.0849 g/s, the total mass of the melted material is *M* = 0.283 + 0.0849 = 0.368 g/s. The thermal diffusivity of the base material is *α* = 9.1 mm^2^/s [6]. The average width of our surface welds is *b_w_* = 8.9 mm and the average cross-sectional area is *A_w_* = 16.5 mm^2^. The kinematic viscosity of the molten steel was calculated from the dynamic viscosity. From Figures 6.17 in [21] and 2.19 in [30], it can be concluded that the average value of the dynamic viscosity of the iron or steel melt just above the melting point is *μ* = 5.4 × 10^3^ Ns/m^2^ (kg/ms), which is too large a value by a factor of 10^6^ (these kinds of mistakes are unacceptable in published materials). This can be concluded from Figure 2.20 in [30] with the correct value *μ* = 5.4 × 10^−3^ kg/ms (g/mm∙s). As the density of the molten steel just above the melting point is *ρ* ≈ 7 × 10^−3^ g/mm^3^ (Figures 6.13a in [21] and 2.21 in [30]), it means that the kinematic viscosity of the steel melt just above the melting point is *ν* = *μ*/*ρ* = 0.77 mm^2^/s. The melting efficiency for our welding conditions calculated with the previously mentioned equations is presented in Table 5.

#### 3.5.2. Determining the Melting Efficiency from the Heat Balance of the Welding Process

The melting efficiency can also be estimated from the known heat balance of the welding process. It is estimated that 40% of all the heat created during arc welding is used to melt the base material [31]. Furthermore, when using a coated electrode, another 30% is used to melt the electrode, with 15% used to melt the metal wire and 15% to melt the electrode coating [8]. When dealing with a coated electrode, it is also necessary to take into account the portions of the metal wire and the coating. Furthermore, it is necessary to take into account that the electrode coating contains metallic alloying components, which are melted and are constituent parts of the surface weld. It is also necessary to take into account that the melting enthalpy of the non-metallic slag components is approximately 10% of the value of the melting enthalpy of the metal. This means that only approximately 10% of the 15% of the heat to melt the electrode coating is consumed for melting the non-metallic components in the coating. The melting efficiency *η_m_*, therefore, consists of partial efficiencies belonging to the base material and the coated electrode. The general equation can be written as:(17)$$\eta_{m}=\eta_{mbm}+\eta_{me}$$
where *η_mbm_* is the melting efficiency of the base material and *η_me_* is the melting efficiency of all the metallic components in the electrode.

It is valid for the admixing rate of the samples that our average surface weld consists of 30% base material and of 70% filler material. This 70% of the surface weld results from the steel wire and from the approximately 40 vol.% of metallic components [32] in the basic coating of the electrodes. Measurements of the diameters of the electrodes showed that the average factor of the coating is ε = 1.5. Due to ε being the ratio between the total electrode diameter and the wire diameter (ε = *Φ_el_*/*Φ_w_*) it follows that 0.5 of that belongs to the coating. The portion belonging to the metal in the coating is 0.5 × 0.4 = 0.2 and the portion belong to the non-metal in the coating is 0.5 × 0.6 = 0.3. The ratio of the metal in the electrode is *ε_m_* = 1 + 0.2 = 1.2, which means that 1.2 × 100/1.5 = 80 vol.% of the electrode belongs to the metal, and from this 20 vol.% to the non-metal. The volume percentage of the components must be converted to a mass percentage to be able to calculate the melting efficiency. For an easier calculation, the density of iron can be used for the metallic part (*ρ_Fe_* = 7.85 g/cm^3^), which means that the factor of the metal mass in 80 vol.% of the electrode is 7.85 × 0.8 ≈ 6.3. The non-metallic part of the basic coating is made of approximately 45% CaCO_3_, 45% CaF_2_ and 10% SiO_2_ [12], the densities of which are: *ρ_CaCO_3__* = 2.77 g/cm^3^, *ρ_CaF_2__* = 3.18 g/cm^3^, and *ρ_SiO_2__* = 2.65 g/cm^3^. The average density of the non-metals in the electrode is (0.45 × 2.77 + 0.45 × 3.18 + 0.1 × 2.65) = 2.94 g/cm^3^, which means that the factor of non-metal mass in 20 vol.% of the electrode is 2.94 × 0.2 ≈ 0.6. The ratio 6.3/0.6 = 10.5 shows that there is 10.5-times more metallic than non-metallic mass in the electrode or that about 90% of the mass of the coated electrode (10.5/11.55 = 0.91) is metal.

The melting efficiency can now be estimated with the following explanation: 30% of the surface weld from the melted base material consumes 40% of the heat, while 70% of the surface weld, which is created from the coated electrode with a 90% metallic part, consumes 30% of the heat. If we also consider the melting enthalpy (*E* of non-metal ≈ 0.1 *E* of metal), approximately 90% of the heat in the coating is consumed for the metallic part. Thus, the melting efficiency from Equation (17) is *η_m_* = (0.3 × 0.4) + (0.7 × 0.9 × 0.3 × 0.9) = 0.29, which is in accordance with the values from the other equations.

Table 5 shows that some equations lead to extremely low values of the melting efficiency for our surface welding conditions, making them unsuitable in our case. Equation (9) is not suitable, as it describes 2D heat dissipation in a completely different geometry of the sample, as well as a different welding technique (Figure 135 in [20] or Figure 3.9 in [21]). Despite discussing single-beaded surface welding, Equation (13a) remains unsuitable, as it is based on completely different welding parameters (much higher currents, voltages, welding speeds and quantities of melted filler material). Equation (16) is also unsuitable as it deals with laser welding.

Of all the equations with the same order of magnitude *η_m_*, the one that gives the lowest value is Rykalin’s Equation (8), which deals with melting of the surface of a thick plate without any filler material. The graphically determined value of the melting efficiency from Figure 134 in [20] is *η_m_* = 0.17 and this was determined from the ratio of the average largest depth *h_max_*, the average width *b_w_* of all our surface welds (in our case *h_max_* ≈ 1.0 mm, *b_w_* = 8.9 mm and *h_max_*/*b_w_* ≈ 0.11) and from dimensionless factors (in our case *ξ*^3*D*^ = 5.86). The lower melting efficiency is a logical consequence of the melted base material only (filler material was not used). This means that according to the known heat balance for arc welding, just 40% of the heat that is used to melt the base material (30% of our surface weld) is taken into account in Figure 134 in [20]. The other 30% of the heat to melt the coated electrode (70% of our surface weld) is not taken into account in Figure 134 because the heat is used to warm the un-melting electrode. Thus, for surfacing, the melting efficiency and the deposition efficiency of the filler material must be added to the graphically determined Rykalin value. Based on this explanation, a new equation was developed:(18)$$\eta_{m}=\eta_{Rykalin}+\left(\eta_{me}\cdot \eta_{dep}\right)$$
where *η_Rykalin_* is the graphically determined Rykalin melting efficiency in the base material, *η_me_* is the melting efficiency of all the metallic components in the electrode and *η_dep_* is the deposition efficiency of the filler metal. If the part belonging to the filler material for the other 70% of our surface weld is added to the value from the diagram, and if we take into account the loss of metallic mass as well (for the basic coated electrode a loss of about 30% was found in Table 10 in [29] and 15–20% in Table 9 in [33]; for the calculation, an average value of 23% was chosen, therefore the average deposition efficiency is *η_dep_* = 77%), the melting efficiency according to Equation (18) is: *η_m_* = 0.17 + (0.7 × 0.9 × 0.3 × 0.9 × 0.77) = 0.30. This result exactly matches with the other values. That is why this value is used for Equation (8) in Table 5.

It is logical that the highest value of *η_m_* is given by Equation (10) for 2D heat flow. It is interesting that of all the comparable equations, Equation (13c) for the 3D heat flow gives a higher value of the melting efficiency than Equation (13b) for the 2D heat flow. This is not in accordance with known facts. This shows how the result is dependent on the chosen constants *η_max_* and *ψ* in Equations (13)–(13c). Due to this, all the suitable values (*η_m_* ≥ 0.27) were taken into account to calculate the average melting efficiency, regardless of the type of heat flow. The average melting efficiency of the arc surface welding with a coated electrode is *η_m_* = 0.3, which shows the excellent choice of value in [26].

By inserting the average value of the melting efficiency, together with the values of the other quantities into Equation (6), we obtain a calculated admixing rate for the S355 steel into the surface welds *D_cal_* = 32%. It matches very well with the already-determined average value, based on experimental methods (the accuracy is ≈ 94%). The calculating test using Equation (6) shows that the melting efficiency *η_m_* = 0.29 gives an admixing rate that is the same as the experimentally determined average admixing rate *D_cal_* = *D_m_* = 30%. The results show the credibility of Equation (6) as well as the correctly determined values of all the material and process quantities required for the calculation. The determined admixing rate also validated the average melting efficiency of the surface arc welding with coated electrodes, which is *η_m_* ≈ 0.30 or, to be more accordant with our results, *η_m_* = 0.29. On the basis of this result a new equation for calculating the melting efficiency based on the welding parameters was developed:(19)$$\eta_{m}=0.346\cdot \exp\left(\frac{-E\cdot \alpha^{2}}{U\cdot I\cdot \eta_{a}\cdot v_{w}}\right)$$

This equation can be used for an estimation of the melting efficiency for the arc welding of a thick or medium-thick plate with a coated electrode. The melting efficiency calculated with Equation (19) using our welding parameters and material constants is *η_m_* = 0.29, which is in accordance with our experimental results.

## 4. Conclusions

Single-bead surface welds were arc surface welded using coated electrodes with different chromium contents onto a 20-mm-thick plate made of S355 steel. The purpose was to determine the admixing rate of the base material into the surface welds. This was done using various methods on a metallographic cross-section of each surface weld. On the basis of the results of the experimental and theoretical research and analyses, we can conclude that:
-The weld pool is very intensely and well mixed during arc surfacing with coated electrodes. This is proven by the very small deviation in the chemical compositions of the different regions in the single-bead surface welds.-The admixing rate of the base material can be determined in various ways: from the ratio of the areas on the cross-section of the surface weld; from the ratio of the heights on the cross-section of the surface weld; with EDS analyses of different areas of the surface weld; and theoretically from the welding parameters, the correctly determined heat and melting efficiency, the materials constants and the dimensions of the weld.-The admixing rate determined from the ratio of the heights is accurate only if there are many measurements made in the direction from the middle towards both edges of the surface weld in each individual cross-section, as the ratio of the heights changes in the same direction. Similarly, in the case of the EDS chemical analyses, it is necessary to analyse the chemical composition of the surface weld cross-section at various points in the region that belongs to the base material and in the region that belongs to the filler material. For each cross-section, it is useful to calculate the admixing rate of all the measured values and of the two lowest values separately, and then finally calculate the average admixing rate of all these values. The results show that this method leads to a very accurately determined admixing rate.-The average admixing rate determined from eight cross-sections and with different experimental methods is the same and equals *D_m_* ≈ 30%, regardless of the deviations in individual surface welds. This means that for an accurate determination of the degree of admixing of the base material in an individual surface weld, regardless of the method used, it is necessary to analyse more metallographic cross-sections, taken from the entire length of the weld at various distances from the start to the end of the weld.-The admixing rate in each surface weld estimated from the ratio of areas was checked and additionally confirmed by comparing the calculated and measured values of chromium in the surface welds. These values match with a 90% to 99% accuracy.-The average melting efficiency of arc surface welding with a coated electrode is *η_m_* = 0.30, and this matches well with the value reported in [26]. The average degree of admixing calculated using Equation (6) with the average melting efficiency *η_m_* = 0.3 and the experimentally used welding parameters and material constants is *D_m_* = 32%, which is 94% accurate with respect to the experimentally measured value.-The experimentally determined admixing rate *D_m_* = 30% shows, from Equation (6), that the melting efficiency in our case of surface welding was *η_m_* = 0.29. Irrespective of the value (*η_m_* = 0.29 or *η_m_* = 0.30), the result shows the credibility of Equation (6) as well as a correct determination of the values of all the material and process quantities used in the calculations.-The melting efficiency can be estimated using various equations on the basis of the welding parameters, the geometry of the weld, the material quantities and the admixing rate. The melting efficiency can also be estimated from a good knowledge of the heat balance of the welding process. It is the sum of the melting efficiency of the base material *η_mbm_* and the melting efficiency of all the metallic components in the electrode *η_me_*: $$\eta_{m}=\eta_{mbm}+\eta_{me}.$$-Rykalin’s method only gives a melting efficiency of 40% for the heat that is used to melt the base material. Therefore, the values of the melting efficiency are low and do not apply to the surface welding with the filler material. If this low value of the melting efficiency *η_Rykalin_* is added to the melting efficiency for all the metallic components in the electrode *η_me_*, calculated from the known heat balance, and if we also take into account the deposition efficiency of the filler metal *η_dep_* during the melting of the coated electrode, we obtain a correct value for the melting efficiency of the surface arc welding of thick and medium-thick plates with the newly developed equation:$$\eta_{m}=\eta_{Rykalin}+\left(\eta_{me}\cdot \eta_{dep}\right).$$-A new equation for the estimation of the melting efficiency of arc welding with a coated electrode from the welding parameters and the material constants was also developed: $$\eta_{m}=0.346\cdot \exp\left(\frac{-E\cdot \alpha^{2}}{U\cdot I\cdot \eta_{a}\cdot v_{w}}\right).$$

## Figures and Tables

**Figure 1 materials-12-01479-f001:**
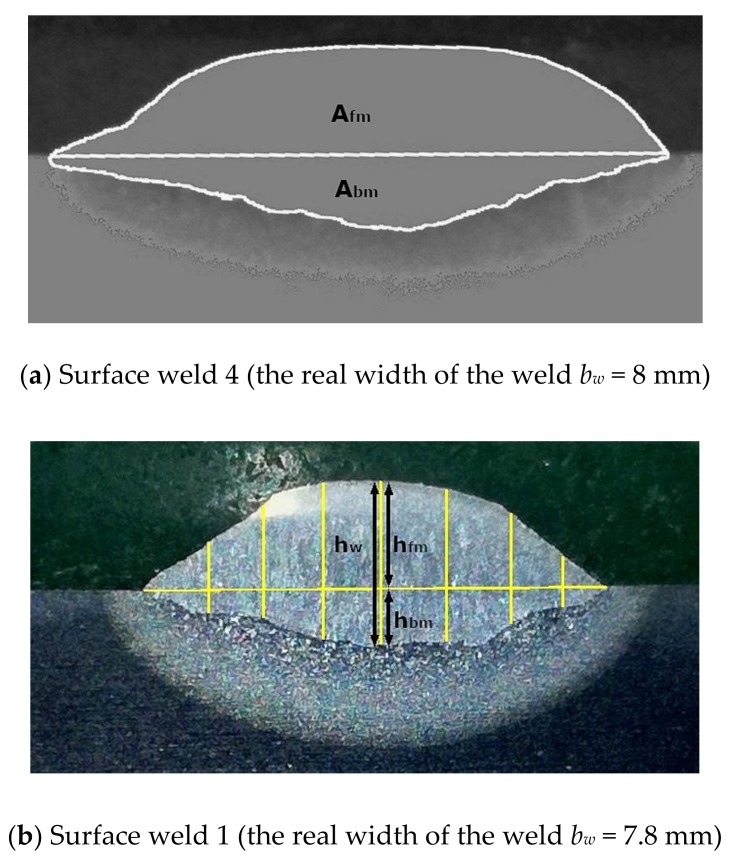
Experimental methods used to determine the admixing rate of the S355 steel into the surface weld: (**a**) from the ratio of the areas; the contrast and outlines of the areas are made with a computer program; (**b**) from the ratio of the heights; (**c**) from the chemical composition of various areas of the surface welds.

**Table 1 materials-12-01479-t001:** Areas of the cross-sections and the admixing rate *D_A_* of the S355 steel in the surface welds.

Weld	1	2	3	4	5	6	7	8
$A_{bm}$ (mm^2^)	4.403	6.424	4.131	3.738	6.755	3.094	4.448	5.983
$A_{w}$ (mm^2^)	13.067	17.875	18.693	12.017	17.990	14.721	11.472	26.427
$D_{A}\;\left(\%\right)$	33.7	35.9	22.1	31.1	37.5	21.0	38.8	22.6

**Table 2 materials-12-01479-t002:** Heights of the cross-sections (the average of seven measurements for individual welds) and the admixing rate *D_h_* of the S355 steel in the surface welds.

Weld	1	2	3	4	5	6	7	8
$h_{bm}$ (mm)	4.775	5.769	3.347	4.164	5.593	2.486	4.680	4.710
$h_{w}$ (mm)	14.452	16.421	15.559	12.806	15.208	11.756	12.344	19.582
$D_{h}\;\left(\%\right)$	33.0	35.1	21.5	32.5	36.8	21.1	37.9	24.0

**Table 3 materials-12-01479-t003:** Contents of the chemical elements Cr, Si, and Mn in the S355 steel and in the surface welds (wt.%) and the admixing rates of the S355 steels *D_A_* and *D^#^_ch_* in the single-bead surface welds (%).

Steel	*Cr*	*Cr*	*Cr*	*Cr*	$Cr_{}^{w}$	$Cr^{fm}/Cr^{bm}$	$D_{A}\;\left(\%\right)$	$Cr^{bc}/Cr^{b*}$	$D_{ch}^{\#}\;\left(\%\right)$	$D_{ch}^{\#}/D_{A}$
S355	$0.42^{Si}$	$1.00^{Mn}$								
$1^{a}$	$0.65^{Si}$	$0.90^{Mn}$								
$1^{fm}$	$0.59^{Si}$	$0.97^{Mn}$								
$1^{bm}$	$0.56^{Si}$	$0.83^{Mn}$				$1.05^{Si},1.17^{Mn}$				
$1^{ev}$	$0.575^{Si}$	$0.90^{Mn}$						$0.99^{Si},\;1.03^{Mn}$		
$1^{bc}$	$0.57^{Si}$	$0.93^{Mn}$					33.7			
$2^{a}$			1.03							
$2^{fm}$	0.63	0.77	0.71	0.68	0.70					
$2^{bm}$	0.73	0.75			0.74	0.946				
$2^{ev}$					0.72			0.917	30.0	0.836
$2^{min}$					0.655			1.008	36.4	1.014
$2^{bc}$					0.66		35.9		33.2	0.925
$3^{a}$			4.55							
$3^{fm}$	3.24	3.33	3.17	3.28	3.255					
$3^{bm}$	3.31	3.42			3.365	0.967				
$3^{ev}$					3.31			1.069	27.2	1.231
$3^{min}$					3.20			1.106	29.7	1.344
$3^{bc}$					3.54		22.1		28.4	1.285
$4^{a}$			9.32							
$4^{fm}$	6.56	6.44	6.52	6.52	6.51					
$4^{bm}$	6.63	6.47			6.55	0.994				
$4^{ev}$					6.53			0.955	29.9	0.903
$4^{min}$					6.455			0.967	30.7	0.927
$4^{bc}$					6.24		31.1		30.3	0.974
$5^{a}$			1.43							
$5^{fm}$	0.88	0.86	0.85	0.94	0.88					
$5^{bm}$	0.91	0.87			0.89	0.989				
$5^{ev}$					0.885			1.006	38.1	1.016
$5^{min}$					0.855			1.041	40.2	1.072
$5^{bc}$					0.89		37.5		39.1	1.043
$6^{a}$			7.10							
$6^{fm}$	6.14	6.17	6.26	6.34	6.23					
$6^{bm}$	5.86	5.39			5.625	1.107				
$6^{ev}$					5.93			0.946	16.5	0.786
$6^{min}$					5.625			0.997	20.8	0.990
$6^{bc}$					5.61		21.0		18.6	0.886
$7^{a}$			7.60							
$7^{fm}$	4.48	4.85	4.28	4.49	4.525					
$7^{bm}$	4.42	4.39			4.405	1.027				
$7^{ev}$					4.465			1.041	41.2	1.062
$7^{min}$					4.335			1.073	42.9	1.106
$7^{bc}$					4.65		38.8		42.0	1.082
$8^{a}$			7.20							
$8^{fm}$	5.54	5.63	5.49	5.48	5.535					
$8^{bm}$	6.07	5.87			5.97	0.927				
$8^{ev}$					5.75			0.967	20.1	0.881
$8^{min}$					5.49			1.013	23.7	1.039
$8^{bc}$					5.56		22.6		21.9	0.969

Marks: *a*—measured content in the filler metal, surface of the three-layered weld; *fm*—measured content in the cross-section of the single-bead surface weld, area belongs to the filler material; *bm*—measured content in the cross-section of the single-bead surface weld, area belongs to the base metal; *ev* = 0.5∙(*fm* + *bm*); *min*—average of the two lowest measured values in the cross-section; *bc*—calculated average value of the chemical element with the admixing rate *D_A_*; Cr*^b*^* is Cr*^ev^* or Cr*^min^* (see the marks near the number of the welds); Cr*^w^*—final average content of Cr; *D^#^_ch_* is *D^t^_ch_*, *D^l^_ch_* or *D_ch_*.

**Table 4 materials-12-01479-t004:** Comparison of the admixing rates determined by different methods.

Weld	1	2	3	4	5	6	7	8	$D_{ev}\;\left(\%\right)$
$D_{A}\;\left(\%\right)$	33.7	35.9	22.1	31.1	37.5	21.0	38.8	22.6	30.3
$D_{h}\;\left(\%\right)$	33.0	35.1	21.5	32.5	36.8	21.1	37.9	24.0	30.2
$D_{ch}\;\left(\%\right)$	/	33.2	28.4	30.3	39.1	18.6	42.0	21.9	30.5

**Table 5 materials-12-01479-t005:** Calculated melting efficiency *η_m_*.

Equation	(8)	(9)	(10)	(11)	(12)	(13a)	(13b)	(13c)	(14)	(15)	(16)	(17)	[26] *
*η_m_*	0.30 ^#^	0.01 ^##^	0.39	0.29	0.32	0.04	0.27	0.34	0.27	0.27	0.07	0.29	0.30

* value from calculated example for arc welding in reference [26]; ^#^ Rykalin’s value *η_m_* = 0.17, corrected with a calculation based on the heat balance (see the text); ^##^
*ξ*^2*D*^ = 0.94 and from the diagram *η_m_* = 0.01.

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
