# Peer review of "Determining the Degree of Admixing Rate of the Base Material and the Melting Efficiency in Single-Bead Surface Welds Using Different Methods, Including New Approaches"

_materials, 2019, doi:10.3390/ma12091479_

Round 1
Reviewer 1 Report
It is aninteresting study about the admixing in WM.
However the reviewer is not a native English speaking person, the language needs serious improvement, there are sentences very hard to understand. Sentences which are longer than 3 lines should be shortened (there is a 14 line long sentence; lines 285-299).
The structure of the paper is very uncommon, in the methods part the methods are not clearly presented but in the R&D....and there are results in the methods section. It needs to be cleared.
Also to give sample calculations for the equations is very uncommon. If the authos feel, the description for equations need to be more clear (i feel the same) describe it more precise! Also more telling abbrevations could help a lot. Also could a comparison of the results of different methods in diagrams.
Also results needs to be shortened at least to its half length a restructured.
Some remarks, questions, comments are also listed in the manuscript_with_reviewers_comments
After restructuring the manuscript might fulfill the publication criteria in Materials

Author Response
Responses to reviewer 1 comments:
a) The paper was proofread by English native speaker dr. Paul McGuiness. The certificate of proofreading is attached.
b) The text was reconstructed as much as possible and some sentences were shortened as it was suggested by the reviewer. Also some parts of the text in the manuscript were improved in order to be more understandable.
We also tried to separate the description of investigating methods from the chapter ˝Results and discussion˝; however, we didn’t succeed completely because of theoretical analysis and many equations. Namely, the present structure of the article enables that the reader does not need to leap from the results to the equations and vice versa. Therefore, all equations and explanations are still presented in chapter ˝Results and discussion˝. This also the reason why this chapter was not shortened as much as the reviewer has proposed.
The comparison of the experimental results was made in Table 4. Due to comparison analyses of results of different methods only column chart would be appropriate. Since there is a lot of results (9 groups with 3 columns = 27 columns) with only slight differences in the values the presentations with the chart would be non-transparent.
c) Regarding the comment which was put in the text in pdf file:
From each weld one cross-section were made. Thus, the average value of the admixing rate was calculated from all 8 cross-sections. If we can assume that 8 different cross-sections of 8 different weld are taken from one single weld it is clear that the average value of admixing rate in a single weld with applied methods will be determined very precisely despite fluctuations of welding conditions.
All modifications in the manuscript are highlighted.
We would like to thank you for your remarks and suggestions which have improved the paper quality.
Hoping that the above mentioned changes in the manuscript and our answers satisfy all reviewer comments, I look forward to hearing from you soon.
Yours sincerely,
Assist. prof. Borut Zorc

Reviewer 2 Report
The article concerns advanced and multifaceted analysis of the degree of mixing the substrate and padding weld. The presented results are very important and can significantly affect the development of knowledge in the field of surface modification and regeneration using welding methods.
Comments:
Language and style should be revised, preferably by a native English speaker.
I suggest taking into account the citations listed below for the needs of the state of the art http://dx.doi.org/10.26628/wtr.v90i10.963
http://dx.doi.org/10.26628/wtr.v90i10.965
The welding method studied is characterized by low repeatability of conditions due to unstable conditions such as welding speed and arc length, they put a shadow on the value of mixing.
The first conclusion, line 407 is obvious and known.
The cited literature is generally old.
Author Response
Responses to reviewer 2 comments:
a) The paper was proofread by English native speaker dr. Paul McGuiness. The certificate of proofreading is attached.
b) The articles proposed by reviewer are proper for the manuscript and they are now cited in the manuscript
c) In generally the repeatability of the welding conditions in our research is not important because the article discuss the different determination methods of admixing rate. The repeatability of the results of admixing rate is important not the repeatability of welding conditions because the article do not discuss the effect of welding conditions on the admixing rate.
If the results of different methods match well it means that admixing rate is well determinate, testing methods are appropriate and the repeatability of the results of the applied methods is guaranteed.
If in 8 different welds which were made with 8 different coated electrodes practically the same results of admixing rate would be obtained, then this may put a shadow on the value of the admixing rate. However, this impossible due to different composition of coating of electrodes. Our results of admixing rate which are in the range of 20 – 40 % confirm your comment about low repeatability conditions of welding. On the other hand, it is well known that in the single weld the welding conditions never fluctuate as much as our results of admixing if it is welded by experienced welder. If we can assume that 8 different cross-sections of 8 different weld are taken from one single weld it is clear that the average value of admixing rate in a single weld with applied methods will be determined very precisely despite fluctuations of welding conditions.
Regarding the comment about the average values of welding parameters: The average values of welding parameters were applied only for the theoretical analyse where for the calculations of the average values are always used. Despite that we put the parameters range in the text.
d) We agree with reviver that first conclusion is already known. However, we only want to present that this fact can be also confirmed with our results on different alloyed surface welds.
e) We generally agree with the reviewer. However, because of welding with coated electrodes some data for the calculations and theoretical analyses were found only in older literature. Anyway, we also additionally cited the proposed literature from the recent date.
f) Because of suggestions of one of the reviewers the article was partly reconstructed. Also new Table (Table 4) was inserted. All modifications in the text are highlighted.
We would like to thank you for your remarks and suggestions which have improved the paper quality.
Hoping that the above mentioned changes in the manuscript and our answers satisfy all reviewer comments, I look forward to hearing from you soon.
Yours sincerely,
Assist. prof. Borut Zorc

Reviewer 3 Report
The following points and questions are presented regarding the manuscript:
The language of the manuscript should be refined; the paper would benefit from some closer proof reading as it includes numerous linguistic errors.
In the Materials and Samples Preparation chapter the authors stated: "Eight steels with different chromium contents (Table 1) were surface welded onto the 20-mm thick, non-preheated, S355 low-alloyed steel sheet using arc welding in a flat position (PA) with commercial, basic coated electrodes with a diameter of 3.25 mm. The welds were single-bead and 150mm long. The average welding parameters were: current I = 100 A, voltage U = 22.5 V, and welding speed Vw = 2.83 mm/s"
From the above description I think that the authors used the Shielded metal arc welding (SMAW) technique, also known as manual metal arc welding (MMA) which is a manual arc welding process that uses a consumable electrode covered with a flux to lay the weld.
Because the MMA method is manual, we can't keep the arc length and welding speed constant, i.e. the parameters are constantly changing, and we can't just report the average values. The properties of the eight commercial basic coated electrodes are not reported. The location of metallographic examinations isn't reported. Under these circumstances how did you prove repeatability of the tests and of the experimental results?
SCALE bars are missing in all Figures.
Author Response
Responses to reviewer 3 comments:
a) The paper was proofread by English native speaker dr. Paul McGuiness. The certificate of proofreading is attached.
b) In generally the repeatability of the welding conditions in our research is not important because the article discuss the different determination methods of admixing rate. The repeatability of the results of admixing rate is important not the repeatability of welding conditions because the article do not discuss the effect of welding conditions on the admixing rate.
If the results of different methods match well it means that admixing rate is well determinate, testing methods are appropriate and the repeatability of the results of the applied methods is guaranteed.
If in 8 different welds which were made with 8 different coated electrodes practically the same results of admixing rate would be obtained, then this may put a shadow on the value of the admixing rate. However, this impossible due to different composition of coating of electrodes. Our results of admixing rate which are in the range of 20 – 40 % confirm your comment about low repeatability conditions of welding. On the other hand, it is well known that in the single weld the welding conditions never fluctuate as much as our results of admixing if it is welded by experienced welder. If we can assume that 8 different cross-sections of 8 different weld are taken from one single weld it is clear that the average value of admixing rate in a single weld with applied methods will be determined very precisely despite fluctuations of welding conditions.
Regarding the comment about the average values of welding parameters: The average values of welding parameters were applied only for the theoretical analyse where for the calculations the average values are always used. Despite that we put the parameters range in the text.
c) Since macroscopic cross-sections of the welds were taken by camera it is very hard to put the exact scale bar in the Figures. Therefore, we rather wrote the real width of the presented welds in figure captions.
d) Because of suggestions of one of the other reviewers the article was partly reconstructed. Also new Table (Table 4) was inserted. All modifications in the text are highlighted.
We would like to thank you for your remarks and suggestions which have improved the paper quality.
Hoping that the above mentioned changes in the manuscript and our answers will satisfy all reviewer comments, I look forward to hearing from you soon.
Yours sincerely,
Assist. prof. Borut Zorc

Round 2
Reviewer 1 Report
The manuscript was improved significantly from the initial state.
Some corrections wouzld be still appropriate.
e.g.:
- a macroimage of the multilayered weld.
- please use reasonable digits for a given measurement, it is virtually impossible to measure a technical weld to 0.001 mm^2 accuracy, and it is also unnecessary, please give 3- standard deviation or scatter for the average values.
- The conclusion is still very long (over 1 page!) and contains discussion...it can be and should be shortened at least by 50 % --> higher infodensity.
Some other remarks and questions are also in the manuscript_with_reviewers_comments (in the attachment).
With proper corrections I think the manuscript could satisfy the publication criteria in Materials.

Author Response
Responses to reviewer 1 comments:
- Multi-layered welds were done only for determination of chemical composition of filler materials which were later necessary for determination of admixing rate from chemical analyses. Thus, metallographic samples of multi-layered welds were not prepared since the purpose of the research was the determination of admixing rate in single-bead surface welds.
- The measurements were obtained by ImageJ software calculations and were not measured by technical methods. The values of the measurements were intentionally on 0.001 since we wanted to calculate admixing precisely as much as possible in order to avoid summarizing the errors in final values of admixing rate. However, the values of calculated admixing rates were then rounded on 0.1. If we rounded the starting values on less digits, the final calculated values would be changed and not so accurate.
- After reading the conclusions many times, we think that these conclusions show the key points and findings. We still believe that short explanation is better than bare listing of conclusions. Since also other two reviewers didn´t have any complaint about the conclusions we would be grateful if we do not need to change them, because we still do not have an idea how to write them in different form.
- The manuscript was proofread twice by English native speaker dr. Paul McGuinness. The certificate of proofreading was sent to the Editor.
Hoping that the above comments satisfy all reviewer remarks, I look forward to hearing from you soon.
Yours sincerely,
Assist. prof. Borut Zorc

Reviewer 3 Report
No further comments
Author Response
Responses to reviewer 3 comments:
We would like to thank you for revision of the manuscript. We also agree that all articles could be always improved or written on different ways due to subjective nature of each author.
The manuscript was proofread twice by English native speaker dr. Paul McGuinness. The certificate of proofreading was sent to the Editor.
Hoping that the above comments satisfy all reviewer remarks, I look forward to hearing from you soon.
Yours sincerely,
Assist. prof. Borut Zorc